# SEMCACHE: ADAPTIVE SEMANTIC-AWARE CACHING FOR EFFICIENT VIDEO DIFFUSION

algorithm algorithmic

## ABSTRACT

Diffusion models have achieved significant progress in video generation tasks, but slow inference speed remains a major challenge. Existing cache-based acceleration methods for video diffusion have demonstrated considerable improvements in inference speed. An existing efficient caching strategy involves reusing model outputs by estimating and leveraging the fluctuating differences among model outputs across timesteps. However, the strategy relies on extensive calibration sets and neglects the fact that prompt semantic variations affect the variational differences in model outputs. This phenomenon is illustrated by comparing videos generated from different prompts: while "a horse running on the grassland" produces highly dynamic content, "a person reading in a coffee shop" results in relatively static scenes. Building on this observation, we propose a novel training-free SemCache method that can adaptively adjust caching strategies by perceiving prompt semantics changes. Key innovations include a Prompt Semantic-Aware (PSA) caching that evaluates prompt semantics and then dynamically decides a caching strategy tailored to the current timestep based on semantic information. We further introduce a Temporal Motion Metric (TMM) strategy to guide the compute allocation along the temporal dimension based on motion information, which not only ensures motion consistency in videos but also further reduces inference time. Experimental results demonstrate that SemCache achieves 2.45× and 2.66× speedups on HunyuanVideo and Wan2.1 respectively, while maintaining high video quality. Our code will be made publicly available.

## 1 INTRODUCTION

Recent developments in diffusion models have demonstrated exceptional performance across visual generation domains. The field has witnessed a paradigm shift from traditional U-Net architectures, toward advanced diffusion transformer frameworks (DiT) (Peebles & Xie, 2023), resulting in substantial improvements in both model expressiveness and output quality. Building upon these innovations, video synthesis models now produce videos with outstanding visual fidelity and consistent temporal dynamics. Despite remarkable progress, diffusion models suffer from prohibitive inference latency, creating a substantial obstacle for real-world applications. This performance bottleneck originates from the step-wise denoising mechanism , which becomes computationally prohibitive when scaling to higher resolutions and longer temporal sequences. Existing acceleration methodologies, such as knowledge distillation (Meng et al., 2023a; Salimans & Ho, 2022a) and quantization approaches (Li et al., 2023c; Chen et al., 2025b; Shang et al., 2023b), present viable solutions but impose significant computational overhead through model retraining requirements and additional data dependencies, constraining their practical utility.

To achieve cost-effective acceleration, training-free caching methods (Li et al., 2024; Wang et al., 2024; Cui et al., 2025) are widely adopted. By reusing intermediate computations across denoising steps, they avoid retraining and calibration, offering practical speedups for pretrained models. Yet most existing approaches make inflexible caching decisions. Static strategies apply fixed reuse patterns, failing to capture dynamic feature similarities. Adaptive methods (Kahatapitiya et al., 2024; Li et al., 2023a; Liu et al., 2025) improve flexibility by reacting to output changes, but rely on low-level signals and ignore high-level semantics. As a result, they often over-cache in dynamic scenes,

causing motion artifacts and quality drops. This issue stems from their insensitivity to prompt content. For example, "a horse galloping" involves rapid changes requiring dense computation, while "a person reading" allows heavy reuse. These semantic differences directly affect optimal caching behavior, yet current methods lack the awareness to adapt accordingly.

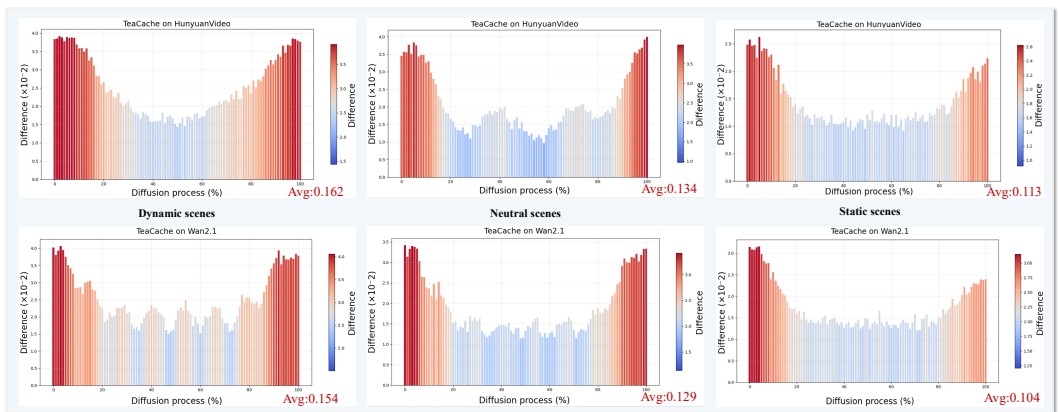

Figure 1: Model output difference comparison during diffusion processes for TeaCache (Liu et al., 2025) on HunyuanVideo and Wan2.1 across static, neutral, and dynamic scenes generated from varying semantic prompts. Scenes are categorized based on input prompts: *static*, *neutral*, and *dynamic*. Higher differences are observed in dynamic scenes, indicating greater feature variation and the need for denser computation.

Prompt semantics naturally capture the global content and dynamics of the target video. To explore their role in caching, we categorize prompts into three types: static, neutral, and dynamic (e.g., "a person reading" vs. "a horse galloping"). When applying existing methods like TeaCache (Liu et al., 2025), we observe that dynamic scenes exhibit larger differences during diffusion process than static ones, as illustrated in Figure 1. This indicates that dynamic content requires denser computation to preserve motion and semantic details, while static scenes allow more aggressive caching due to high frame-to-frame similarity. This suggests a simple but effective principle: caching should adapt to the semantic complexity and motion level implied by the prompt. Instead of relying on low-level pixel changes, we use the prompt as a high-level guide to decide when and what to cache.

Based on this insight, we propose **SemCache**, a training-free semantic-aware caching framework. We introduce a Prompt Semantic-Aware (PSA) strategy that estimates semantic consistency by measuring the magnitude of cross-attention output differences. This lightweight estimation reflects how strongly the model's latent features respond to the text prompt, serving as a proxy for scene complexity. Higher variation indicates dynamic content, leading to less caching; lower variation allows more reuse. To further reduce temporal redundancy, we design a Temporal Motion Metric (TMM) that computes motion intensity from inter-frame latent differences. Together, PSA and TMM generate semantic and motion scores that guide adaptive caching decisions, such as skipping frames or reusing model outputs, without retraining or calibration. Experiments show that SemCache achieves significant speedups (up to 2.66×) while maintaining high visual quality, outperforming existing methods across diverse prompts and models.

Our contributions are summarized as follows:

- We propose **SemCache**, a novel semantic-aware and adaptive caching framework for video diffusion models. It is training-free, general, and leverages prompt semantics to guide efficient computation without retraining or calibration.

- We introduce two key components in SemCache: **Prompt Semantic-Aware (PSA)** and **Temporal Motion Metric (TMM)**. PSA estimates scene complexity from cross-attention output differences to adaptively control spatial and step-level caching; TMM measures latent frame differences to guide temporal computation. Together, they enable content-aware acceleration aligned with both semantics and motion dynamics.

- We evaluate SemCache on multiple video diffusion models (e.g., HunyuanVideo, Wan2.1), achieving up to 2.66× speedup with minimal quality loss, outperforming state-of-the-art methods across diverse static and dynamic scenes.

## 2 RELATED WORKS

### 2.1 DIFFUSION MODELS FOR VIDEO GENERATION

Diffusion models for video generation (Wang et al., 2025; Kong et al., 2024) have demonstrated impressive capabilities in generating high-quality videos from text prompts, with good temporal consistency and visual fidelity. Initially, text-to-video generation was achieved by incorporating temporal fusion modules into existing text-to-image models (Ramesh et al., 2022; Chen et al., 2023; Wei et al., 2023; Rombach et al., 2022; Wang et al., 2023), which showed promising performance in video synthesis. However, when complex scene generation became a requirement, these U-Net-based architectures exhibited significant limitations and poor performance.

To address these shortcomings, Diffusion Transformers (DiT) (Peebles & Xie, 2023) were introduced. Leveraging the scalable architecture of transformers, DiT demonstrates tremendous potential in video generation capabilities. For instance, Open-Sora (Zheng et al., 2024b) can produce videos where objects and scenes follow natural physical laws and realistic motion patterns. Nevertheless, transformer-based diffusion models require several minutes to generate even short-duration videos, making slow inference speed a major bottleneck for real-world applications.

### 2.2 DIFFUSION MODEL ACCELERATION

To address the high computational cost, acceleration methods for diffusion models have been proposed.Existing acceleration methods can be broadly categorized into two categories. First, DDIM (Song et al., 2020) leverages a deterministic sampling process to reduce sampling steps while produce high-quality samples. methods based on step-distillation (Meng et al., 2023b; Salimans & Ho, 2022b) and adaptive high-order solvers (Lu et al., 2023) have been introduced for fast sampling. Additionally, quantization (Li et al., 2023b; Shang et al., 2023a; He et al., 2023; Chen et al., 2025a), and pruning (Zhang et al., 2024) for diffusion models have been explored. However, these approaches often require extra compute overhead. Second, various efforts have been devoted to caching-based methods.

### 2.3 CACHE MECHANISM

cache-based methods has emerged as a particularly promising training-free acceleration paradigm, exploiting the observation that consecutive denoising steps often produce similar intermediate representations. Early caching methods like DeepCache (Xu et al., 2018), Faster Diffusion (Li et al., 2023a) employed uniform strategies, reusing computations at fixed intervals. While simple to implement, these approaches only implemented on the UNet architecture. Recently,PAB (Zhao et al., 2025) reduce Redundant computation by reusing features from DiT block at select timesteps. $\Delta$-DiT (Chen et al., 2024) focused on caching residuals between attention layers. However, these methods still rely on heuristic rules that may not capture the true temporal dynamics of feature evolution.These approaches exhibit promising results, which benefits from heuristic or data-driven patterns.The limitation of static caching schemes has led to increased interest in adaptive approaches that can dynamically adjust their behavior based on runtime characteristics. AdaCache (Kahatapitiya et al., 2024) and Fora (Selvaraju et al., 2024) proposed adaptive routers to optimize reuse per step, while FasterCache (Li et al., 2023a) exploited redundancy in classifier-free guidance outputs. Tea-Cache (Liu et al., 2025) attain speedup by means of a timestep embedding aware mechanism .Other advancements include token caching technologies such as ToCa (Zou et al., 2025) that mitigates computational redundancy in token-level. Although these methods demonstrate promising acceleration performance, they neglect the inherent correlation between prompt semantics and caching strategies.

## 3 METHODOLOGY

### 3.1 PRELIMINARIES

**Diffusion models** generate visual content by employing a series of progressive denoising process. The underlying mechanism starts from random noise and gradually transforming it through successive refinements to produce a sample that matches the target data distribution. The forward diffusion phase involves progressively corrupting an original data point $x_0$ from the real distribution $q(x)$ by gradually introducing Gaussian noise over T steps. Given noise variance schedule $\beta_t$ ,the posterior distribution for the reverse denoising process, $p_\theta(x_{t-1}|x_t)$, This can be formulated as:

$$\mathcal{N}\left(x_{t-1}; \frac{1}{\sqrt{\alpha_t}}\left(x_t - \frac{1-\alpha_t}{\sqrt{1-\bar{\alpha}_t}}\epsilon_\theta(x_t, t)\right), \beta_t\mathbf{I}\right) \tag{1}$$

Here, $\alpha_t = 1 - \beta_t$, $\bar{\alpha}_t = \prod_{i=1}^{T}\alpha_i$, and $T$ denotes the total timestep count. The denoising network $\epsilon_\theta$, parameterized by $\theta$, accepts $x_t$ and $t$ as inputs to estimate the noise required for denoising. For complete image synthesis across $T$ timesteps, $\epsilon_\theta$ must execute $T$ forward passes, which serves as the primary computational bottleneck in diffusion models. Recent research demonstrates that transformer-based implementations of $\epsilon_\theta$ frequently yield superior generation quality.

Diffusion transformers (DiT) are constructed by repeatedly stacking self-attention, MLP, and Cross-Attention layers (for conditional generation) that map the semantics of prompts. It can be roughly formulated as:

$$\mathcal{G} = g_1 \circ g_2 \circ \cdots \circ g_I, \quad \text{where}$$
$$g_l = \mathcal{F}_{SA}^i \circ \mathcal{F}_{CA}^i \circ \mathcal{F}_{MLP}^i \tag{2}$$

$\mathcal{G}$, $g_i$, and $\mathcal{F}^i$ represent the DiT model, the blocks in DiT, and different layers in a DiT block, respectively. $i$ denotes,where the subscript represents the DiT block index and $I$ indicates the total depth of the DiT architecture. For diffusion transformers, the input $\epsilon_t$ is formulated as a token sequence derived from image patches: $\epsilon_t = \{x_i\}_{i=1}^{H \times W}$, with $H$ and $W$ denoting the spatial dimensions of images or their latent representations.

Token caching (Zou et al., 2025) is designed to leverage previously computed and stored features from earlier timesteps, thereby avoiding redundant computations in subsequent timesteps. Additionally, $\delta$-DiT (Chen et al., 2024) involves using feature offsets rather than the feature maps themselves as cache objects to avoid information loss.

### 3.2 ANALYSIS

Diffusion models provide semantic guidance to the model through prompts, enabling the generation of corresponding video content. Different prompts contain varying semantic information; for instance, some prompts require generating complex dynamic scenes while others call for relatively static scenes. Based on the dynamic characteristics of the generated video content, we can roughly categorize them into three types: dynamic, static, and neutral. We observe that dynamic scenes exhibit larger differences during diffusion process than static ones, and this variation in differences is associated with prompt semantics. For prompts requiring complex dynamic scene generation, under the premise of balancing inference speed and video quality, computations should be performed as extensively as possible. For static scene generation, enhanced model output reuse frequency is essential to minimize computational overhead. The current challenge lies in how to implement caching strategies based on semantic information.

A reasonable solution is to score prompt semantics, then adaptively select optimal caching based on prompt semantic scores. Unfortunately, these semantic scores are difficult to obtain. Large Language Models (LLMs) are readily available and possess strong logical reasoning capabilities, therefore we consider leveraging LLMs to score prompt semantics. We select ChatGPT-4o as the semantic scoring model and provide it with knowledge of the relationship between fully computed timesteps for TeaCache and MagCache methods across 10 prompts each for dynamic, neutral, and

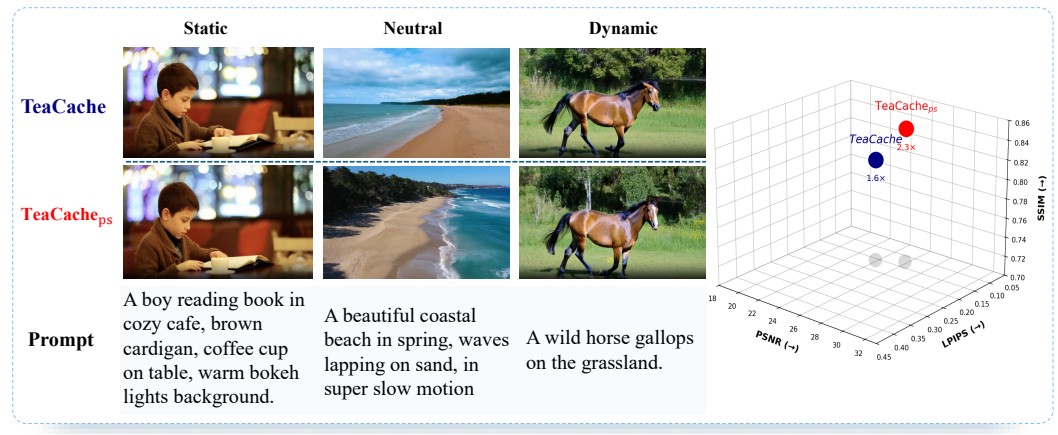

Figure 2: Comparison between TeaCache and TeaCache with prompt semantic scoring on visual quality.

static scene generation. We choose ChatGPT-4o as the semantic scoring model and equip it with background knowledge regarding the timestep computation patterns of TeaCache and MagCache across 30 prompts (10 each for dynamic, neutral, and static scenes). We integrate this strategy into TeaCache (Liu et al., 2025) to assess model performance with respect to inference speed and video generation quality. Experimental results demonstrate that considering prompt semantics can further improve inference speed without compromising video quality, as shown in Fig. 2. To validate the reliability of LLM-based prompt semantic scoring, we perform consistency analysis comparing LLM-generated semantic scores with human annotations under identical conditions. The consistency analysis reveals remarkably high similarity between LLM and human assessments. Detailed experimental results can be found in Appendix A.1. However, introducing LLMs for prompt semantic scoring during the generation process would incur additional costs. The ideal approach is to find a low-cost alternative strategy.

Cross-Attention layers are responsible for interacting with conditions, and given access to attention maps, they can be utilized to obtain useful information related to the generated video content, which can help determine the type of video being generated. However, this approach is extremely time-consuming. We find that prompt semantic scores can be approximately estimated by measuring the strength of cross-attention output differences.

### 3.3 SEMCACHE

Building on the analysis in the above insights, we propose SemCache, a novel training-free acceleration method that introduces a Prompt Semantic-Aware strategy in the spatial dimension to adaptively select optimal caching. Furthermore, we introduce a Temporal Motion Metric (TMM) strategy in the temporal dimension to focus on motion-rich frames and reduce redundant frames. SemCache enables inference acceleration without compromising video quality, as shown in Fig. 3.

**Prompt Semantic-Aware (PSA)** As discussed in Section 3.2, prompt semantics naturally capture the global content and dynamics of the target video. Our analysis reveals that dynamic scenes exhibit larger output variations during the diffusion process than static scenes. This observation leads to a key insight: dynamic scenes require increased computational allocation due to higher variation, while static scenes can benefit from enhanced reuse strategies due to lower variation. Based on this understanding, we introduce a Prompt Semantic-Aware strategy to adaptively select optimal caching by leveraging prompt semantic information.

We initially leverage large language models to score prompt semantics for caching guidance and validate the effectiveness of this approach. However, this method introduces additional computational overhead. To address this limitation, we explore a more efficient alternative. Cross-attention layers in diffusion models facilitate semantic interaction, fundamentally determining the generation

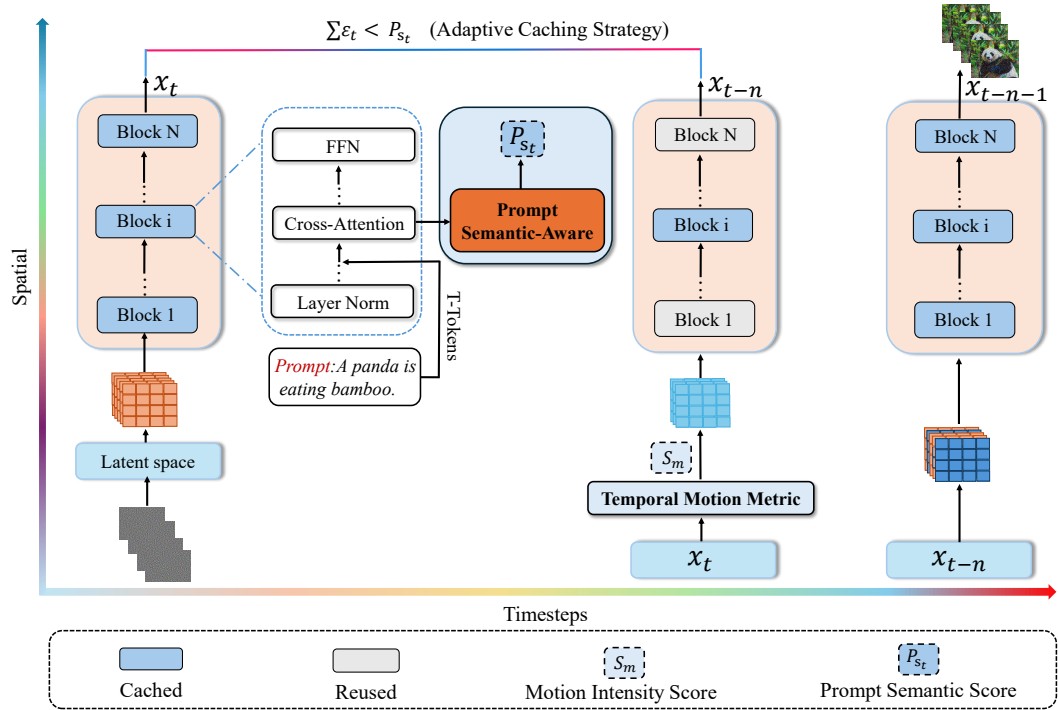

Figure 3: Overview of SemCache. SemCache incorporates a Prompt Semantic-Aware (PSA) strategy and a Temporal Motion Metric (TMM) strategy. The PSA strategy perceives prompt semantics to enable adaptive optimal caching selection in the spatial dimension. The TMM strategy captures motion semantics to prioritize motion-rich frames while minimizing redundant frames in the temporal dimension.

of semantically aligned content. We discover that prompt semantic scores can be approximately estimated by measuring the magnitude of cross-attention output differences. This approach leverages the model's intrinsic characteristics, constituting an extremely low-cost approximation method. More discussion and visualization can be found in Appendix. The process involves calculating the magnitude $M_{raw}$ by computing input-output changes across five layers each from shallow, middle, and deep layers during complete computation timesteps, followed by Tanh normalization, and calculating and normalizing directional variations $D_{norm}$.

$$M_{\text{raw}} = \frac{||\tilde{\mathbf{Z}}_{\text{out}}||_F - ||\tilde{\mathbf{Z}}_{\text{in}}||_F}{||\tilde{\mathbf{Z}}_{\text{in}}||_F + \epsilon} \tag{3}$$

$$M_{\text{norm}} = \tanh\left(\frac{M_{\text{raw}}}{s_m}\right) \tag{4}$$

where: $\tilde{\mathbf{Z}}_{\text{out}}$ and $\tilde{\mathbf{Z}}_{\text{in}}$ denote the averaged cross-attention output and input, respectively. $||\cdot||_F$ denotes the Frobenius norm $\epsilon = 10^{-8}$ ensures numerical stability by preventing division by zero. $s_m$ is the default scaling parameter controlling the tanh saturation rate.

$$\cos\theta = \frac{\langle\tilde{\mathbf{Z}}_{\text{in}}, \tilde{\mathbf{Z}}_{\text{out}}\rangle}{||\tilde{\mathbf{Z}}_{\text{in}}||_F||\tilde{\mathbf{Z}}_{\text{out}}||_F} \tag{5}$$

$$D_{\text{norm}} = \frac{1 - \cos\theta}{2} \tag{6}$$

$$p_{s_t} = \alpha \cdot M_{norm} + \beta \cdot D_{norm} \tag{7}$$

where $\alpha$ and $\beta$ are the weighting hyperparameters for magnitude and direction changes, respectively.

During the warm-up phase, the first timestep contains significant noise, and abnormal values would compromise the accurate computation of $p_{s_t}$. Consequently, $p_{s_t}$ calculation commences from the second timestep. According to Eq. 7, the prompt semantic score $p_{s_t}$ is obtained through weighted fusion of the normalized magnitude $M_{norm}$ and directional variation $D_{norm}$.

SemCache employs the prompt semantic score as $p_{s_t}$ the caching error threshold to enable dynamic adaptation of caching strategies. When the cumulative error exceeds the threshold, it is reset to zero. We maintain a running total error $\mathcal{E}_t$:

$$\mathcal{E}_t = \mathcal{E}_{t-1} + \frac{\|O_t - I_t\|_1}{\|O_{t-1} - I_{t-1}\|_1} \cdot \frac{\|O_t - O_{t-1}\|_1}{\|I_t - I_{t-1}\|_1} \tag{8}$$

Let $t-1$ be the timestep where we refresh the cache, with $O_{t-1}$ and $I_{t-1}$ representing the model output and input at timestep $t-1$, respectively. At timestep $t-1$, the cumulative error $\mathcal{E}_{t-1}$ is initialized to zero. At timestep $t$, since $\mathcal{E}_t < p_{s_t}$ is satisfied, the cache is reused, while $\mathcal{E}_t$ is updated according to Eq. 8.

### 3.4 TEMPORAL MOTION METRIC

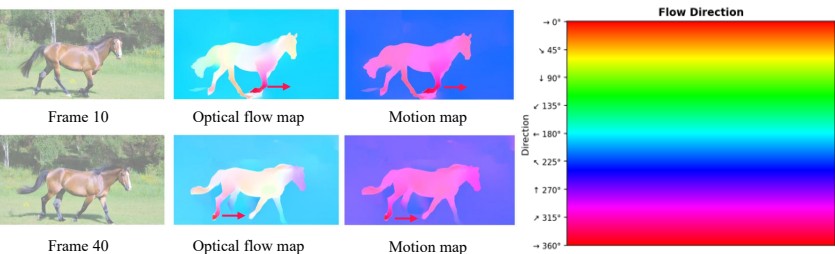

Figure 4: Comparison of motion estimation using optical flow networks versus motion estimation using inter-frame differences.

Maintaining temporal consistency of video content is crucial in video generation. If excessive caching is employed during the video generation process, the resulting videos may suffer from artifacts and other quality issues, while insufficient caching leads to slower inference speed in video generation. As shown in Fig. 4, by using optical flow networks (Teed & Deng, 2020) to perform motion estimation on generated video frames, we found that computing only frames rich in motion information can reduce temporal redundancy while improving inference speed.

**Temporal Motion Metric** To mitigate redundant computations in the video temporal dimension, we introduce a Temporal Motion Metric strategy. Although optical flow networks provide effective video motion estimation, they incur increased computational overhead. For efficient video motion estimation, We compute the residuals between video frames in the latent space to derive Motion Intensity Scores that measure the strength of motion information. which direct temporal dimension caching strategies.

Motion Score Computation is described by the following equation: Given a video sequence with $T$ frames, motion scores are $\{s_1, s_2, \ldots, s_{T-1}\}$. The Motion Intensity Score $S_m$ for $f$-th frame is calculated as:

$$S_m = \frac{\|f(t) - f(t-1)\|_2}{\|f(t)\|_2 + \|f(t-1)\|_2 + \epsilon}, \quad t = 2, 3, \ldots, T \tag{9}$$

where $f(t)$ is the latent space feature vector of frame $t$, $\|\cdot\|_2$ denotes the L2 norm, and $\epsilon$ is a small constant to prevent division by zero.

For target preservation ratio $r$ (e.g., $r = 0.2$ means preserving 20% of frames), the percentile threshold is defined as:

$$\tau = P_{(1-r)\times 100}(\{s_1, s_2, \ldots, s_{T-1}\}) \tag{10}$$

where $P_p(\cdot)$ denotes the $p$-th percentile function. During the generation process, computation is allocated to frames exceeding threshold $\tau$, while frames below threshold $\tau$ are reused. This approach effectively reduces temporal redundancy while ensuring temporal consistency in generated videos.

The SemCache method performs spatial dimension caching by perceiving prompt semantics and guides temporal dimension caching through motion semantics. The combination of both approaches reduces spatial and temporal redundancy, thereby achieving superior acceleration performance.

## 4 EXPERIMENTS

### 4.1 COMPARISON EXPERIMENT

To assess the effectiveness of our method, we apply our acceleration technique to HunyuanVideo (Kong et al., 2024), WAN 2.1 (Wang et al., 2025) and Open-Sora 1.2 (Zheng et al., 2024a). We compare our acceleration algorithm with recent efficient video synthesis techniques, including Tea-Cache (Liu et al., 2025) and MagCache (Ma et al., 2025).

Table 1 reports the quantitative performance of algorithm efficiency and visual quality on the VBench benchmark (Huang et al., 2024).

Table 1: Quantitative evaluation of inference efficiency and visual quality in video generation models. SemCache consistently achieves superior efficiency and better visual quality across different base models, sampling schedulers, video resolutions, and lengths.

| Method | Efficiency | | | Visual Quality | | | |
|---|---|---|---|---|---|---|---|
| | FLOPs (P) ↓ | Speedup ↑ | Latency (s) ↓ | VBench ↑ | LPIPS ↓ | SSIM ↑ | PSNR ↑ |
| **Open-Sora 1.2** (51 frames, 480P) | | | | | | | |
| Open-Sora 1.2 ($T = 30$) | 3.15 | 1× | 44.69 | 0.79.25 | - | - | - |
| PAB-slow | 2.56 | 1.19× | 37.55 | 0.76.11 | 0.2694 | 0.7001 | 19.56 |
| PAB-fast | 2.50 | 1.32× | 33.86 | 0.73.02 | 0.392 | 0.641 | 17.12 |
| TeaCache-fast | 1.64 | 2.07× | 21.58 | 0.78.42 | 0.2529 | 0.7469 | 19.06 |
| MagCache | 1.64 | 2.09× | 21.38 | - | 0.1649 | 0.8106 | 22.15 |
| SemCache | **1.61** | **2.1×** | **21.28** | **0.7843** | **0.1610** | **0.8127** | **22.21** |
| **HunyuanVideo** (129 frames, 1280×720) | | | | | | | |
| HunyuanVideo ($T = 50$) | 27.43 | 1× | 2602.73 | 0.773 | - | - | - |
| TeaCache-slow | 19.17 | 1.71× | 1517.02 | 0.7548 | 0.1787 | 0.7520 | 21.55 |
| TeaCache-fast | 14.89 | 2.05× | 1269.62 | 0.7407 | 0.1891 | 0.7322 | 20.97 |
| MagCache | 13.43 | 2.12× | 1227.7 | - | 0.1726 | 0.7501 | 21.48 |
| SemCache | **12.07** | **2.45×** | **1062.34** | **0.7598** | **0.1664** | **0.7516** | **21.57** |
| **Wan 2.1** (81frames, 832×480) | | | | | | | |
| Wan 2.1 ($T = 50$) | 8.32 | 1× | 364.13 | 0.7275 | - | - | - |
| TeaCache-slow | 5.34 | 1.6× | 227.58 | 0.7122 | 0.1853 | 0.7511 | **23.10** |
| TeaCache-fast | 3.96 | 2.11× | 172.57 | 0.6901 | 0.201 | 0.7344 | 17.92 |
| MagCache | 3.28 | 2.4× | 151.72 | - | 0.1864 | 0.742 | 22.45 |
| SemCache | **3.12** | **2.66×** | **136.9** | **0.7123** | **0.1851** | **0.7513** | 22.97 |

### 4.2 EVALUATION METRICS

To validate the effectiveness of our proposed method, we perform comprehensive experiments across two key dimensions: (1) visual quality evaluation and (2) efficiency evaluation. For visual quality evaluation, we employ VBench (Huang et al., 2024), LPIPS (Zhang et al., 2018), PSNR, and SSIM as evaluation metrics. For efficiency evaluation, we measure FLOPs and latency.

### 4.3 IMPLEMENTATION DETAIL

All experiments are conducted on NVIDIA A100 80GB GPUs using PyTorch, with FlashAttention enabled by default. All experiments follow the default settings of the baseline models. We set the hyperparameters $\alpha$ and $\beta$ in Eq. 7 to ($\alpha = 0.4$, $\beta = 0.6$) for HunyuanVideo, ($\alpha = 0.2$, $\beta = 0.8$) for Wan2.1, and Open-Sora 1.2 ($\alpha = 0.2$, $\beta = 0.8$) respectively.

## 4.4 ABLATION STUDY

We conducted ablation experiments on PSA and TMM to validate their effectiveness. These experiments are performed using Wan 2.1 as the baseline model. We further evaluate the sensitivity to hyperparameters $\alpha$, $\beta$, and the number of cross-attention layers on HunyuanVideo and Wan 2.1.

**Evaluation of PSA.** We set up ablation experiments using only the magnitude changes or directional variations of Cross-Attention input-output to guide spatial caching. As shown in Table 2, we separately tested the caching effectiveness of using only magnitude changes or only directional changes of Cross-Attention input-output on efficiency and visual quality metrics, and compared them with SemCache without the Temporal Motion Metric (TMM). The experimental results demonstrate that combining both magnitude and directional changes of Cross-Attention input-output can efficiently achieve spatial dimension caching.

Table 2: Ablation study of PSA on visual metrics and inference efficiency. 'CM': the magnitude changes of Cross-Attention input-output. 'CD': the directional changes of Cross-Attention input-output.

| Method | FLOPs ↓ | Speedup ↑ | Latency ↑ | VBench ↑ | LPIPS ↓ | SSIM ↑ | PSNR ↑ |
|---|---|---|---|---|---|---|---|
| Wan 2.1 | 8.32 | 1× | 364.13 | 0.7275 | - | - | - |
| Wan 2.1(w/CM) | 6.13 | 1.4× | 260 | 0.7106 | 0.1865 | 0.7588 | 22.96 |
| Wan 2.1(w/CD) | 6.04 | 1.41× | 258.24 | 0.7121 | 0.1892 | 0.7548 | 22.87 |
| SemCache(w/o TMM) | 4.15 | 2.12 × | 171.75 | 0.7104 | 0.1843 | 0.7513 | 22.88 |

**Effect of TMM.** As shown in Table 3, performing only spatial dimension caching without introducing the Temporal Motion Metric (TMM) strategy achieves reasonable performance in efficiency and visual quality metrics. However, when utilizing the Temporal Motion Metric strategy, optimal performance can be achieved, realizing an effective trade-off between video quality and inference speed.

Table 3: Ablation study of TMM on visual quality and inference efficiency

| Method | FLOPs ↓ | Speedup ↑ | Latency ↑ | VBench ↑ | LPIPS ↓ | SSIM ↑ | PSNR ↑ |
|---|---|---|---|---|---|---|---|
| Wan 2.1 | 0.7275 | - | - | - | | | |
| SemCache(w/o TMM) | 4.15 | 2.12× | 171.75 | 0.7104 | **0.1843** | 0.7513 | 22.88 |
| SemCache | **3.12** | **2.66×** | **136.9** | 0.7123 | 0.1851 | **0.7513** | **22.97** |

**Effect of hyperparameters.** To better illustrate the parameter sensitivity of $\alpha$ and $\beta$, we conducted a series of ablation studies. As shown in Figure 5, the Pareto frontier curve reveals a non-linear performance trade-off. Specifically, optimal performance is achieved at $\alpha = 0.4$ for HunyuanVideo and $\alpha = 0.2$ for Wan2.1, yielding significant inference speedup while maintaining high visual quality (VBench scores) and effectively outperforming other configurations along the Pareto frontier.

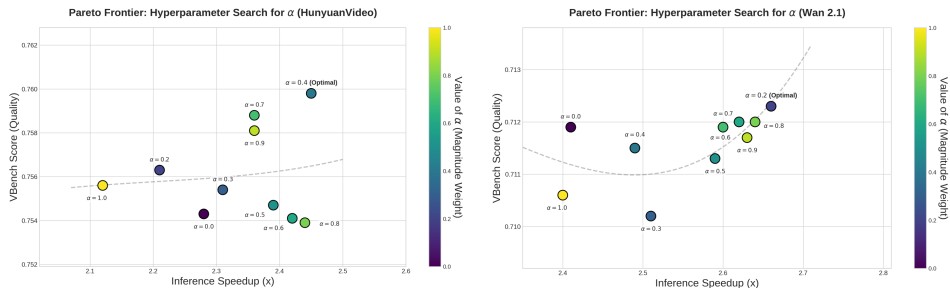

Figure 5: Ablation study on hyperparameter $\alpha$ under the constraint $\alpha + \beta = 1$. Grid search results for HunyuanVideo (left) and Wan 2.1 (right) showing the trade-off between speedup and VBench scores. The Pareto frontier reveals optimal values at $\alpha = 0.4$ and $\alpha = 0.2$ respectively.

**Performance at Different Lengths and Resolutions.** To assess the scalability of our method across different video configurations, we evaluate performance on varying resolutions and frame counts. As shown in Fig. 6, our method achieves consistent acceleration across all tested settings, delivering

approximately 2.2-2.52× speedup on single GPU and 3.0-3.25× speedup on dual GPUs. The stable performance across different resolutions (544P×960P to 720P×1280P) and frame counts (129 to 192 frames) demonstrates the method's robustness.

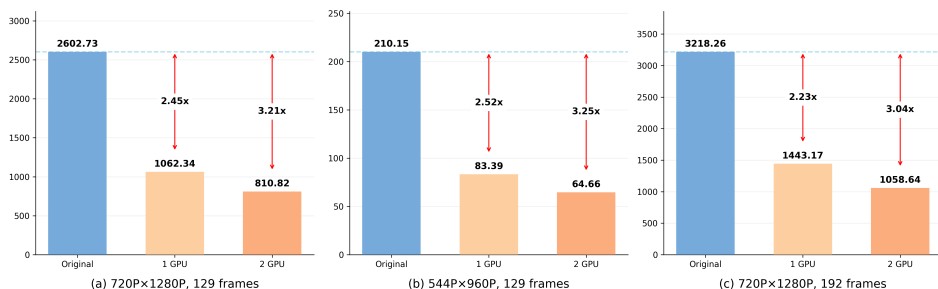

Figure 6: Inference efficiency of SemCache at different video lengths and resolutions.

Table 4 presents the performance sensitivity to the number of cross-attention layers. Results indicate that optimal inference-quality trade-offs are achieved with 12 layers for Open-Sora 1.2, and 15 layers for both HunyuanVideo and Wan 2.1.

Table 4: Ablation study on the number of cross-attention layers for HunyuanVideo, Wan 2.1, and Open-Sora 1.2 models.

| Method | Efficiency | | | Visual Quality | | | |
|---|---|---|---|---|---|---|---|
| | FLOPs (P) ↓ | Speedup ↑ | Latency (s) ↓ | VBench ↑ | LPIPS ↓ | SSIM ↑ | PSNR ↑ |
| **Open-Sora 1.2** (51 frames, 480P) | | | | | | | |
| Open-Sora 1.2 ($T = 30$) | 3.21 | 1× | 44.69 | 0.7925 | - | - | - |
| SemCache (l=3) | 1.52 | 2.41× | 18.54 | 0.7401 | 0.1740 | 0.6546 | 18.24 |
| SemCache (l=6) | 1.54 | 2.33× | 19.18 | 0.7467 | 0.1673 | 0.6814 | 19.17 |
| SemCache (l=9) | 1.57 | 2.28× | 19.6 | 0.7514 | 0.1620 | 0.6943 | 19.68 |
| SemCache (l=12) | **1.61** | **2.1×** | **21.28** | **0.7843** | **0.1610** | **0.8127** | **22.21** |
| SemCache (l=15) | 1.64 | 2.06× | 21.69 | 0.7823 | 0.1579 | 0.7958 | 21.01 |
| SemCache (l=18) | 1.68 | 1.92× | 23.28 | 0.7814 | 0.1533 | 0.8027 | 21.19 |
| **HunyuanVideo** (129 frames, 1280×720) | | | | | | | |
| HunyuanVideo ($T = 50$) | 27.43 | 1× | 2602.73 | 0.773 | - | - | - |
| SemCache ($l = 3$) | 11.35 | 2.62× | 993.40 | 0.7241 | 0.1922 | 0.7244 | 19.12 |
| SemCache ($l = 6$) | 11.48 | 2.57× | 1012.74 | 0.7323 | 0.1810 | 0.7325 | 19.67 |
| SemCache ($l = 9$) | 11.69 | 2.54× | 1024.69 | 0.7409 | 0.1765 | 0.7397 | 20.22 |
| SemCache ($l = 12$) | 11.87 | 2.49× | 1045.27 | 0.7525 | 0.1706 | 0.7433 | 20.97 |
| SemCache ($l = 15$) | **12.07** | **2.45×** | **1062.34** | **0.7598** | **0.1664** | **0.7516** | **21.57** |
| SemCache ($l = 18$) | 12.32 | 2.40× | 1084.47 | 0.7583 | 0.1642 | 0.7554 | 21.19 |
| **Wan 2.1** (81 frames, 832×480) | | | | | | | |
| Wan 2.1 ($T = 50$) | 8.32 | 1× | 364.13 | 0.7275 | - | - | - |
| SemCache ($l = 3$) | 2.79 | 2.81× | 129.58 | 0.6807 | 0.2167 | 0.7086 | 19.37 |
| SemCache ($l = 6$) | 2.84 | 2.76× | 131.93 | 0.6895 | 0.2074 | 0.7159 | 20.46 |
| SemCache ($l = 9$) | 2.91 | 2.72× | 133.87 | 0.7057 | 0.2013 | 0.7281 | 21.03 |
| SemCache ($l = 12$) | 3.07 | 2.69× | 135.36 | 0.7094 | 0.1939 | 0.7353 | 21.88 |
| SemCache ($l = 15$) | **3.12** | **2.66×** | **136.90** | **0.7123** | **0.1851** | **0.7513** | **22.97** |
| SemCache ($l = 18$) | 3.18 | 2.58× | 141.14 | 0.7116 | 0.1834 | 0.7484 | 22.42 |

## 5 CONCLUSION

In this paper, we introduce a novel training-free **SemCache** method featuring a **Prompt Semantic-Aware (PSA)** strategy that estimates semantic consistency by measuring the magnitude of cross-attention output differences. This lightweight estimation reflects how strongly the model's latent features respond to the text prompt, serving as a proxy for scene complexity. Higher variation indicates dynamic content, leading to less caching; lower variation allows more reuse. To further reduce temporal redundancy, we design a **Temporal Motion Metric (TMM)** that computes motion intensity from inter-frame latent differences. Together, PSA and TMM generate semantic and motion scores that guide adaptive caching decisions, without retraining or calibration.

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

# A APPENDIX

## A.1 SEMANTIC SCORING

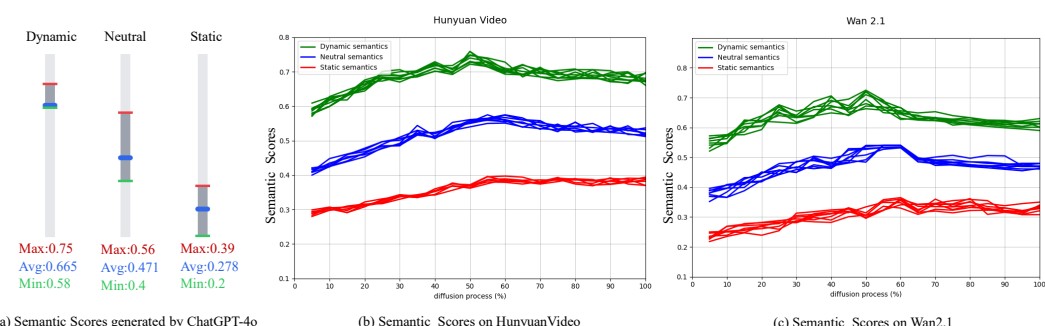

(a) Semantic Scores generated by ChatGPT-4o  (b) Semantic Scores on HunyuanVideo  (c) Semantic Scores on Wan2.1

Figure 7: (a) Statistical analysis of semantic scoring using ChatGPT-4o to evaluate prompts from VBench (Huang et al., 2024) across four dimensions: motion intensity, scene complexity, temporal dynamics, and visual dynamics. (b) and (c) Semantic scoring variation trends during generation for 30 prompts containing dynamic, neutral, and static semantics on HunyuanVideo and Wan2.1, respectively.

We utilized ChatGPT-4o to conduct a comprehensive evaluation of 946 prompts from VBench (Huang et al., 2024) analyzing them across four key aspects: motion intensity, scene complexity, temporal dynamics, and visual dynamics. The corresponding evaluation results are illustrated in Fig. 7 (a). When the prompt semantics involve generating complex dynamic scenes, the semantic score will be higher; conversely, for static or simple scenes, the score will be lower.

## A.2 INFERENCE LATENCY COMPARISON

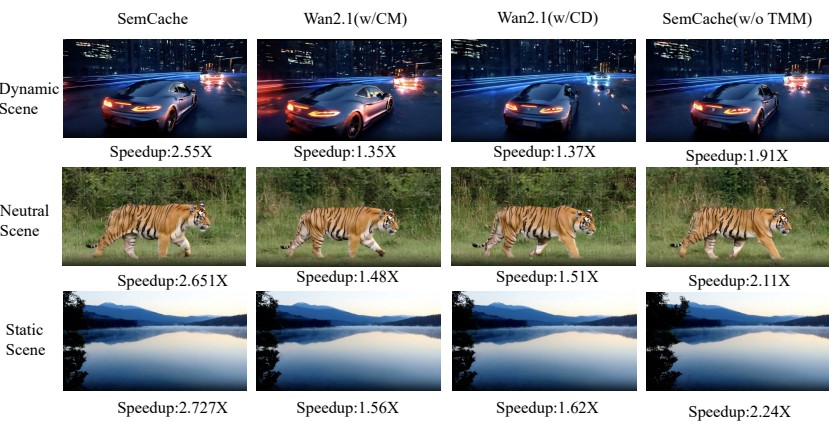

Figure 8: Efficiency comparison (speedup) across three different module configurations: magnitude-only Cross-Attention input-output changes, direction-only changes, and without Temporal Motion Metric (TMM) strategy. SemCache outperforms all alternative settings.

We conduct visualization experiments to compare PSA and TMM strategies, as shown in Fig. 8. Through comparative analysis of inference speeds, SemCache demonstrates superior performance while preserving video generation quality, as shown in Fig. 9.

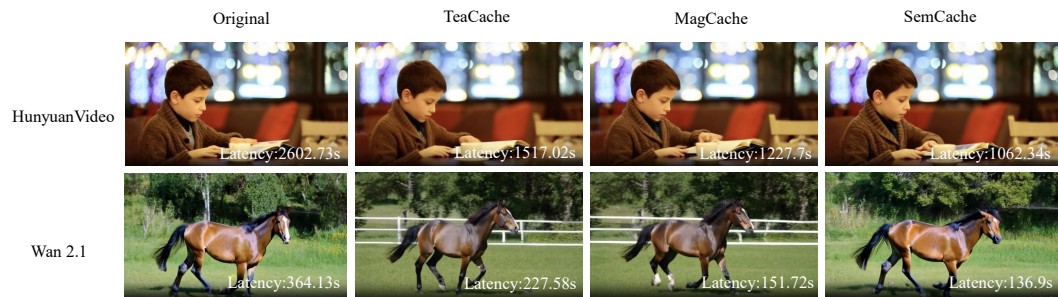

Figure 9: Comparison of inference speed with competing methods. On both HunyuanVideo (Kong et al., 2024) and Wan2.1 (Wang et al., 2025) baselines, SemCache achieves the lowest latency compared to TeaCache and MagCache methods, while maintaining generation quality close to the baseline. This demonstrates that SemCache achieves a favorable trade-off between video quality and inference speed.

## B   PSEUDO CODE OF SEMCACHE

To enhance the reproducibility and clarity of our method, we provide detailed pseudo-code for the core inference strategy of SemCache.

---
**Algorithm 1** SemCache Inference
---

**Require:** Latents $x_T$, Prompt $P$, Total Steps $T$, Warm-up $T_{warm}$, Hyperparams $\alpha, \beta, \tau$
**Ensure:** Generated video $x_0$
1: Init $\mathcal{E} = 0$, Cache $O_{ref} = $ None, $\Delta_{ref} = $ None
2: **for** $t = T$ down to 0 **do**
3:    **if** $t > (T - T_{warm})$ **then**
4:       $O_t \leftarrow \text{Model}(x_t, t, P)$
5:       Update Cache: $O_{ref} \leftarrow O_t$, Reset $\mathcal{E} \leftarrow 0$
6:    **else**
7:       Calculate motion scores $S_m$ for $x_t$ (Eq. 9)
8:       Identify $\mathcal{F}_{static} = \{f \mid S_m < \tau\}$
9:       Compute $M_{norm}, D_{norm}$ from sampled Cross-Attn
10:      Set Threshold $P_{s_t} \leftarrow \alpha \cdot M_{norm} + \beta \cdot D_{norm}$ (Eq. 7)
11:      **for** each frame $f$ in $x_t$ **do**
12:        **if** $f \in \mathcal{F}_{static}$ **then**
13:          $O_t^f \leftarrow \text{Reuse}(x_t^f)$ {TMM}
14:        **else**
15:          Compute output $O_t$
16:          **if** $\mathcal{E} < P_{s_t}$ **then**
17:            **Cache:** $O_t^f \leftarrow O_{ref}^f + \Delta_{ref}^f$ {PSA}
18:            $\mathcal{E} \leftarrow \mathcal{E} + \frac{\|O_t - I_t\|_1}{\|O_{t-1} - I_{t-1}\|_1} \cdot \frac{\|O_t - O_{t-1}\|_1}{\|I_t - I_{t-1}\|_1}$ (Eq. 8)
19:          **else**
20:            **Cache Miss:** $O_t^f \leftarrow \text{Model}(x_t^f, t, P)$
21:            Update Cache: $O_{ref}^f \leftarrow O_t^f$
22:            Reset $\mathcal{E} \leftarrow 0$
23:          **end if**
24:        **end if**
25:      **end for**
26:    **end if**
27:    $x_{t-1} \leftarrow \text{Scheduler}(x_t, O_t)$
28: **end for**
29: **return** $x_0$

---

## C    ETHICS STATEMENT

This work adheres to the ICLR Code of Ethics. In this study, no human subjects or animal experimentation was involved. We have taken care to avoid any biases or discriminatory outcomes in our research process. No personally identifiable information was used, and no experiments were conducted that could raise privacy or security concerns. We are committed to maintaining transparency and integrity throughout the research process.

## D    REPRODUCIBILITY STATEMENT

We have made every effort to ensure that the results presented in this paper are reproducible. All code and datasets have been made publicly available in an anonymous repository to facilitate replication and verification. The experimental setup, including training steps, model configurations, and hardware details, is described in detail in the paper. We provide a comprehensive description of our novel semantic-aware and adaptive caching framework for video diffusion models to facilitate reproduction of our experiments.

We believe these measures will enable other researchers to reproduce our work and further advance the field.

## E    LLM USAGE

Large Language Models (LLMs) were used to aid in the writing and polishing of the manuscript. Specifically, we used an LLM to assist in refining the language, improving readability, and ensuring clarity in various sections of the paper. The model helped with tasks such as sentence rephrasing, grammar checking, and enhancing the overall flow of the text.

It is important to note that the LLM was not involved in the ideation, research methodology, or experimental design. All research concepts, ideas, and analyses were developed and conducted by the authors. The contributions of the LLM were solely focused on improving the linguistic quality of the paper, with no involvement in the scientific content or data analysis.

The authors take full responsibility for the content of the manuscript, including any text generated or polished by the LLM. We have ensured that the LLM-generated text adheres to ethical guidelines and does not contribute to plagiarism or scientific misconduct.

