# OpenReview forum: "SemCache:Adaptive Semantic-Aware Caching for Efficient Video Diffusion"
_ICLR.cc/2026/Conference — Submitted to ICLR 2026_

### Official Review · Reviewer_6Jrv · 2025-10-29

**Soundness:** 3
**Presentation:** 2
**Contribution:** 2
**Rating:** 4
**Confidence:** 3

**Summary:**

This paper introduces SemCache, a training-free semantic-aware caching framework that accelerates video diffusion inference by adapting caching strategies to prompt semantics and motion dynamics. It combines Prompt Semantic-Aware (PSA) caching, which estimates scene complexity via cross-attention variations, and Temporal Motion Metric (TMM), which measures latent motion intensity to guide computation allocation.

**Strengths:**

1. This paper  uses prompt semantics and motion perception to guide caching in diffusion transformers, bridging high-level text understanding and low-level feature reuse.

2. In the exeperiments part, it demonstrates  speedup and high-quality preservation across two large-scale video diffusion backbones and multiple metrics.

**Weaknesses:**

1.In the experimental section, several important details are missing, and the experiments do not adequately highlight or validate the main motivation of the paper (see detailed comments 1–3).

2.The connection between cross-attention variation and semantic complexity, while intuitive, lacks a rigorous analytical foundation or sensitivity analysis.


Detailed comments:

1. In the experimental section, several details are unclear. For instance, for the FLOPs metric, it is not specified whether the reported values are in terabytes, gigabytes, or another unit. Additionally, key implementation details such as the generated video length and the prompt size are not provided.

2. More importantly, the paper is motivated by the observation that prompts with different semantic complexities can influence caching behavior. However, in the experimental results, the authors do not explicitly evaluate performance(not only limit to latency) under prompts with diverse semantic dynamics (e.g., static vs. dynamic scenes). As a result, it is difficult to verify whether the proposed semantic-aware mechanism truly adapts to varying prompt semantics as claimed.


3. Table 1 is not placed correctly in the paper.

4. In the early example where an LLM is used to analyze prompt semantic scoring, it is unclear how the method determines which cache entries to retain, since all prompts are static. Would the model then keep all cached results at every step?

**Questions:**

See above

---

> ### Author Response · Authors · 2025-12-04
> **Response to Reviewer 6Jrv**
>
> ## W1: FLOPs metric etc
>
> We sincerely thank the reviewers for their time and insightful feedback.
>
> We apologize for any oversight.
>
> Video Length: As stated in the header of Table 1, the generated videos are 129 frames (1280x720) for HunyuanVideo, 51 frames (848x480) for Open-Sora 1.2  and 81 frames (832x480) for Wan 2.1.
>
>  FLOPs Unit: The reported values in Table 1 are in Peta-FLOPs ($10^{15}$), representing the total computational cost for generating a single video.
>
> Prompt Details: We utilized standard prompts from the VBench benchmark, adhering to the default tokenizer limits of the respective models.
>
> ## W2 and W4: Prompt semantic scoring
>
>  To better demonstrate that this proxy correlates strongly with computational redundancy, we conduct comparative experiments using 100 prompts for dynamic scenes,  and 100 prompts for static scenes, comparing a uniform acceleration strategy against SemCache. As shown in the table below, experimental results reveal that applying the same acceleration ratio to dynamic videos as to static videos leads to significant degradation in video generation quality.
>
> For static video generation, the model exhibits low sensitivity to caching errors, allowing more aggressive caching to reduce redundant computation. In contrast, for dynamic video generation, the model is highly sensitive to caching errors; excessive caching causes error accumulation, increasing the probability of artifacts in generated dynamic videos. SemCache dynamically perceives prompt semantics to achieve optimal trade-offs between acceleration and generation quality. These findings validate that this proxy correlates strongly with computational redundancy.
> | Method | Speedup $\uparrow$ | LPIPS $\downarrow$ | SSIM $\uparrow$ | PSNR $\uparrow$ |
> | :--- | :---: | :---: | :---: | :---: |
> | &nbsp;&nbsp;&nbsp;&nbsp;&nbsp;&nbsp;&nbsp;&nbsp;&nbsp;&nbsp;&nbsp;&nbsp;&nbsp;&nbsp;&nbsp;&nbsp;&nbsp;&nbsp;&nbsp;&nbsp;&nbsp;&nbsp;&nbsp;&nbsp;&nbsp;&nbsp;&nbsp;&nbsp;&nbsp;&nbsp;&nbsp;&nbsp;&nbsp;&nbsp;&nbsp;&nbsp; **Open-Sora 1.2 (51 frames, 480P)** | | | | |
> | SemCache (Static scenes w/o PSA) | 2.1$\times$ | 0.1715 | 0.7809 | 21.58 |
> | SemCache (Static scenes) | **2.1$\times$** | **0.1236** | **0.8206** | **25.84** |
> | SemCache (Dynamic scenes w/o PSA) | 2.1$\times$ | 0.2910 | 0.7103 | 17.16 |
> | SemCache (Dynamic scenes) | **2.03$\times$** | **0.1397** | **0.8128** | **23.39** |
> | &nbsp;&nbsp;&nbsp;&nbsp;&nbsp;&nbsp;&nbsp;&nbsp;&nbsp;&nbsp;&nbsp;&nbsp;&nbsp;&nbsp;&nbsp;&nbsp;&nbsp;&nbsp;&nbsp;&nbsp;&nbsp;&nbsp;&nbsp;&nbsp;&nbsp;&nbsp;&nbsp;&nbsp;&nbsp;&nbsp;&nbsp;&nbsp;&nbsp;&nbsp;&nbsp;&nbsp; **HunyuanVideo (129 frames, 1280$\times$720)** | | | | |
> | SemCache (Static scenes w/o PSA) | 2.45$\times$ | 0.195 | 0.7439 | 21.76 |
> | SemCache (Static scenes) | **2.45$\times$** | **0.1502** | **0.7819** | **24.98** |
> | SemCache (Dynamic scenes w/o PSA) | 2.45$\times$ | 0.2110 | 0.7118 | 18.22 |
> | SemCache (Dynamic scenes) | **2.38$\times$** | **0.1783** | **0.7405** | **21.06** |
> | &nbsp;&nbsp;&nbsp;&nbsp;&nbsp;&nbsp;&nbsp;&nbsp;&nbsp;&nbsp;&nbsp;&nbsp;&nbsp;&nbsp;&nbsp;&nbsp;&nbsp;&nbsp;&nbsp;&nbsp;&nbsp;&nbsp;&nbsp;&nbsp;&nbsp;&nbsp;&nbsp;&nbsp;&nbsp;&nbsp;&nbsp;&nbsp;&nbsp;&nbsp;&nbsp;&nbsp; **Wan 2.1 (81 frames, 832$\times$480)** | | | | |
> | SemCache (Static scenes w/o PSA) | 2.66$\times$ | 0.1763 | 0.7351 | 21.45 |
> | SemCache (Static scenes) | **2.66$\times$** | **0.1694** | **0.7604** | **23.78** |
> | SemCache (Dynamic scenes w/o PSA) | 2.66$\times$ | 0.2064 | 0.7252 | 17.06 |
> | SemCache (Dynamic scenes) | **2.49$\times$** | **0.1864** | **0.7481** | **21.76** |

---

### Official Review · Reviewer_GAqx · 2025-10-31

**Soundness:** 2
**Presentation:** 3
**Contribution:** 2
**Rating:** 4
**Confidence:** 3

**Summary:**

This paper proposes to accelerate the inference speed of video diffusion models by addressing a key limitation in existing cache-based methods: they largely neglect how prompt semantic variations (e.g., "a horse running" vs. "a person reading") affect model output differences and, consequently, the optimal caching strategy. To tackle this, the authors introduce SemCache, a novel, training-free method that adaptively adjusts its caching strategy by perceiving prompt semantics. Building on the observation that dynamic scenes exhibit larger output variations than static ones during the diffusion process, SemCache introduces two key components. First, a Prompt Semantic-Aware (PSA) strategy estimates semantic consistency by measuring the magnitude of cross-attention output differences, where higher variation indicates dynamic content and leads to less caching, while lower variation allows for more reuse. Second, a Temporal Motion Metric (TMM) strategy computes motion intensity from inter-frame latent differences to guide compute allocation along the temporal dimension, thereby reducing redundancy while ensuring motion consistency. Experimental results demonstrate that SemCache achieves significant speedups (up to 2.45x on Hunyuan Video and 2.66x on Wan2.1) while maintaining high video quality.

**Strengths:**

* The Prompt Semantic-Aware (PSA) strategy is a novel and interesting idea, using cross-attention output differences as a lightweight, training-free method to guide model reuse and computation.
* The TMM method is built on the highly intuitive and logical premise that dynamic and static scenes require different computational loads, and it uses this insight to further improve the efficiency of the diffusion process.
* The experimental section validates the reliability of the proposed method by comparing it directly against recent SOTA acceleration methods, including TeaCache and MagCache. The results show that SemCache achieves superior efficiency while maintaining competitive visual quality.

**Weaknesses:**

* Unclear Methodology Visualization (Figure 3): Figure 3 is ambiguous. The horizontal "Temporal" axis label conflicts with the visual examples ($x_t$, $x_{t-n}$), which represent diffusion timesteps. The diagram also fails to clearly illustrate how PSA guides module reuse or how TMM differentially allocates resources.
* Logical Contradiction in PSA Error Metric: The paper's logic for the error metric Equation 8 is critically flawed. It requires the current output $O_t$ to decide whether to compute $O_t4$, creating a circular dependency.
The required low-cost, predictive error estimation (implied by $\Sigma\epsilon_t < P_{s_t}$) is not explained.
* Lack of clarity in TMM and PSA integration: The paper fails to clearly describe or illustrate the integration between TMM and PSA. It does not specify how the two systems interact or how TMM’s frame filtering mechanism aligns with PSA’s step filtering, leaving readers to infer the operational relationship on their own.

**Questions:**

* The TMM's "skip frame" mechanism seems absolute. Was a "smoother" approach considered?

---

> ### Author Response · Authors · 2025-12-04
> **Response to Reviewer GAqx**
>
> ## W2:  Logic for the error metric Equation 8
> We sincerely thank the reviewers for their time and insightful feedback
>
> We respectfully clarify that there is **no circular dependency**. The decision logic is **Pre-hoc** (based on history), while the update is **Post-hoc**.
>
> * **Step $t$ Decision:** We check if the accumulated error from previous steps ($ {t+1}$) is less than threshold  $P_{s_t}$.
>     * **If Yes:** We cache. We approximate $O_t$.
>     * **If No:** We compute $O_t$ fully.
>
> * **Step $t$ Update:** After obtaining $O_t$ (via approximation or computation), we calculate the difference (Eq. 8) and add it to the accumulator for the next step decision.
>
> Thus, we do not need $O_t$ to decide for step $t$; we rely on the accumulated risk from $t+1, \dots, T$.
>
> ## W3: TMM and PSA integration
> We sincerely thank the reviewers for their time and insightful feedback. We apologize for any confusion regarding SemCache.
> We clarify the complete pipeline of SemCache as follows:
> * **What is cached**: SemCache caches the model block outputs (specifically, the output-input difference $\Delta$) from fully computed timesteps, rather than caching raw intermediate features.
>
> * **When to cache**: The caching decision is dynamically determined by our PSA metric—when cumulative error $E_t < P_{st}$ (semantic threshold), we reuse cached outputs; otherwise, we perform full computation and update the cache.
>
> * **Where it operates**: Caching occurs at the Transformer block level, meaning the decision applies to complete block forward passes, not individual layers or attention heads.
> We have formalized this in Algorithm 1 (Appendix B) with explicit pseudocode showing the cache/compute decision logic.
>
> | Method | Speedup $\uparrow$ | LPIPS $\downarrow$ | SSIM $\uparrow$ | PSNR $\uparrow$ |
> | :--- | :---: | :---: | :---: | :---: |
> | &nbsp;&nbsp;&nbsp;&nbsp;&nbsp;&nbsp;&nbsp;&nbsp;&nbsp;&nbsp;&nbsp;&nbsp;&nbsp;&nbsp;&nbsp;&nbsp;&nbsp;&nbsp;&nbsp;&nbsp;&nbsp;&nbsp;&nbsp;&nbsp;&nbsp;&nbsp;&nbsp;&nbsp;&nbsp;&nbsp;&nbsp;&nbsp;&nbsp;&nbsp;&nbsp;&nbsp; **Open-Sora 1.2 (51 frames, 480P)** | | | | |
> | SemCache (Static scenes w/o PSA) | 2.1$\times$ | 0.1715 | 0.7809 | 21.58 |
> | SemCache (Static scenes) | **2.1$\times$** | **0.1236** | **0.8206** | **25.84** |
> | SemCache (Dynamic scenes w/o PSA) | 2.1$\times$ | 0.2910 | 0.7103 | 17.16 |
> | SemCache (Dynamic scenes) | **2.03$\times$** | **0.1397** | **0.8128** | **23.39** |
> | &nbsp;&nbsp;&nbsp;&nbsp;&nbsp;&nbsp;&nbsp;&nbsp;&nbsp;&nbsp;&nbsp;&nbsp;&nbsp;&nbsp;&nbsp;&nbsp;&nbsp;&nbsp;&nbsp;&nbsp;&nbsp;&nbsp;&nbsp;&nbsp;&nbsp;&nbsp;&nbsp;&nbsp;&nbsp;&nbsp;&nbsp;&nbsp;&nbsp;&nbsp;&nbsp;&nbsp; **HunyuanVideo (129 frames, 1280$\times$720)** | | | | |
> | SemCache (Static scenes w/o PSA) | 2.45$\times$ | 0.195 | 0.7439 | 21.76 |
> | SemCache (Static scenes) | **2.45$\times$** | **0.1502** | **0.7819** | **24.98** |
> | SemCache (Dynamic scenes w/o PSA) | 2.45$\times$ | 0.2110 | 0.7118 | 18.22 |
> | SemCache (Dynamic scenes) | **2.38$\times$** | **0.1783** | **0.7405** | **21.06** |
> | &nbsp;&nbsp;&nbsp;&nbsp;&nbsp;&nbsp;&nbsp;&nbsp;&nbsp;&nbsp;&nbsp;&nbsp;&nbsp;&nbsp;&nbsp;&nbsp;&nbsp;&nbsp;&nbsp;&nbsp;&nbsp;&nbsp;&nbsp;&nbsp;&nbsp;&nbsp;&nbsp;&nbsp;&nbsp;&nbsp;&nbsp;&nbsp;&nbsp;&nbsp;&nbsp;&nbsp; **Wan 2.1 (81 frames, 832$\times$480)** | | | | |
> | SemCache (Static scenes w/o PSA) | 2.66$\times$ | 0.1763 | 0.7351 | 21.45 |
> | SemCache (Static scenes) | **2.66$\times$** | **0.1694** | **0.7604** | **23.78** |
> | SemCache (Dynamic scenes w/o PSA) | 2.66$\times$ | 0.2064 | 0.7252 | 17.06 |
> | SemCache (Dynamic scenes) | **2.49$\times$** | **0.1864** | **0.7481** | **21.76** |

---

### Official Review · Reviewer_tWL2 · 2025-11-01

**Soundness:** 3
**Presentation:** 2
**Contribution:** 2
**Rating:** 4
**Confidence:** 3

**Summary:**

This paper introduces SemCache, a training-free acceleration framework for video diffusion models. Its primary novelty is the introduction of "semantic awareness" to caching. The method is motivated by the insight that static scenes are more compressible than dynamic scenes. SemCache uses a dual strategy: (1) a Prompt Semantic-Aware (PSA) module that uses changes in cross-attention outputs as a lightweight proxy for scene complexity to guide caching across diffusion steps, and (2) a Temporal Motion Metric (TMM) that uses latent frame differences to guide caching across time. Experiments on Hunyuan Video and Wan 2.1 show state-of-the-art speedups (up to 2.66x) while maintaining high visual quality, outperforming other caching baselines.

**Strengths:**

1. The core idea of linking caching strategy to the prompt's semantic content (dynamic vs. static) is a clear and logical advancement over existing methods that are "blind" to this high-level context.
2. The paper's most clever contribution is the PSA module. The authors first prove that an LLM *could* provide this semantic guidance, but then successfully replace this costly component with a lightweight and intrinsic signal: the magnitude and directional changes in cross-attention outputs. This is a well-designed solution in general.

**Weaknesses:**

1. The method's robustness is questionable due to key hyperparameters that appear to be model-specific. The weights for the PSA score ($\alpha$ and $\beta$) are set to (0.4, 0.6) for Hunyuan Video but (0.2, 0.8) for Wan 2.1. The paper provides no methodology for how these weights were found, suggesting a process of manual, model-specific tuning that may not generalize easily to new models.
2. The method is only validated on two video diffusion models. While these are large, recent models, competitors (like BWCache in the previous review) were tested on a wider array of five models. This limited scope makes it difficult to assess the true generality of the approach.
3. The PSA score is calculated using cross-attention outputs from "five layers each from shallow, middle, and deep layers". This selection (15 specific layers) seems arbitrary and is not justified. It is unclear how these layers were chosen or how sensitive the method's performance is to this specific selection, which weakens the robustness of the proposed proxy metric.
4. The paper shows that using an LLM works (Figure 2) and that its proxy works (Table 2). However, it never directly compares them. It would be much stronger if it showed that the proxy-generated scores ($p_{s_t}$) are highly correlated with the LLM-generated scores. This would provide direct evidence that the proxy is *actually* capturing "semantic complexity" as claimed.

**Questions:**

Have the authors evaluated the computational overhead of calculating PSA and TMM scores, especially for real-time generation scenarios?

---

> ### Author Response · Authors · 2025-12-04
> **Response to Reviewer tWL2**
>
> ## W1 and W2: Ask for hyperparameters
> We sincerely thank the reviewers for their time and insightful feedback.
>
> To better evaluate the performance impact of $\alpha$ and $\beta$ in the new model, we benchmarked OpenSora 1.2 using the configurations from HunyuanVideo ($\alpha=0.4$) and Wan2.1 ($\alpha=2$). As shown in the table below, even when adopting the same hyperparameters as Wan2.1, the model achieves a favorable trade-off between generation speed and quality.
>
>
>
> | Method | FLOPs (P) $\downarrow$ | Speedup $\uparrow$ | Latency (s) $\downarrow$ | VBench $\uparrow$ | LPIPS $\downarrow$ | SSIM $\uparrow$ | PSNR $\uparrow$ |
> | :--- | :---: | :---: | :---: | :---: | :---: | :---: | :---: |
> | Open-Sora 1.2 ($T=30$) | 3.15 | 1$\times$ | 44.69 | 0.7925 | - | - | - |
> | PAB-slow | 2.56 | 1.19 $\times$ | 37.55 | 0.7611 | 0.2694 | 0.7001 | 19.56 |
> | PAB-fast | 2.50 | 1.32$\times$ | 33.86 | 0.7302 | 0.392 | 0.641 | 17.12 |
> | TeaCache-fast | 1.64 | 2.07$\times$ | 21.58 | 0.7842 | 0.2529 | 0.7469 | 19.06 |
> | MagCache | 1.64 | 2.09$\times$ | 21.38 | - | 0.1649 | 0.8106 | 22.15 |
> | SemCache($\alpha=0.4, \beta=0.6$) | 1.63 | 2.09$\times$ | 21.38 | 0.7842 | 0.1621 | 0.8109 | 22.14 |
> | SemCache($\alpha=0.2, \beta=0.8$) | **1.61** | **2.1$\times$** | **21.28** | **0.7843** | **0.1610** | **0.8127** | **22.21** |
>
> ## W3: Selection of number of layers
>
> We thank the reviewer for this question.
>
> We conducted comparative experiments on the number of cross-attention layers, and the results are presented in the table below. The experimental results indicate that while an insufficient number of layers yields a higher speed-up ratio, it results in significantly degraded generation quality. Conversely, an excessive number of layers negatively impacts generation speed.
>
>
> **(a) Open-Sora 1.2 (51 frames, 480P)**
>
> | Method | FLOPs (P) ↓ | Speedup ↑ | Latency (s) ↓ | VBench ↑ | LPIPS ↓ | SSIM ↑ | PSNR ↑ |
> | :--- | :---: | :---: | :---: | :---: | :---: | :---: | :---: |
> | Open-Sora 1.2 (T=30) | 3.21 | 1× | 44.69 | 0.7925 | - | - | - |
> | SemCache ($l$=3) | 1.52 | 2.41× | 18.54 | 0.7401 | 0.1740 | 0.6546 | 18.24 |
> | SemCache ($l$=6) | 1.54 | 2.33× | 19.18 | 0.7467 | 0.1673 | 0.6814 | 19.17 |
> | SemCache ($l$=9) | 1.57 | 2.28× | 19.60 | 0.7514 | 0.1620 | 0.6943 | 19.68 |
> | **SemCache ($l$=12)** | **1.61** | **2.1×** | **21.28** | **0.7843** | **0.1610** | **0.8127** | **22.21** |
> | SemCache ($l$=15) | 1.64 | 2.06× | 21.69 | 0.7823 | 0.1579 | 0.7958 | 21.01 |
> | SemCache ($l$=18) | 1.68 | 1.92× | 23.28 | 0.7814 | 0.1533 | 0.8027 | 21.19 |
>
> **(b) HunyuanVideo (129 frames, 1280×720)**
>
> | Method | FLOPs (P) ↓ | Speedup ↑ | Latency (s) ↓ | VBench ↑ | LPIPS ↓ | SSIM ↑ | PSNR ↑ |
> | :--- | :---: | :---: | :---: | :---: | :---: | :---: | :---: |
> | HunyuanVideo (T=50) | 27.43 | 1× | 2602.73 | 0.773 | - | - | - |
> | SemCache ($l$=3) | 11.35 | 2.62× | 993.40 | 0.7241 | 0.1922 | 0.7244 | 19.12 |
> | SemCache ($l$=6) | 11.48 | 2.57× | 1012.74 | 0.7323 | 0.1810 | 0.7325 | 19.67 |
> | SemCache ($l$=9) | 11.69 | 2.54× | 1024.69 | 0.7409 | 0.1765 | 0.7397 | 20.22 |
> | SemCache ($l$=12) | 11.87 | 2.49× | 1045.27 | 0.7525 | 0.1706 | 0.7433 | 20.97 |
> | **SemCache ($l$=15)** | **12.07** | **2.45×** | **1062.34** | **0.7598** | **0.1664** | **0.7516** | **21.57** |
> | SemCache ($l$=18) | 12.32 | 2.40× | 1084.47 | 0.7583 | 0.1642 | 0.7554 | 21.19 |
>
> **(c) Wan 2.1 (81 frames, 832×480)**
>
> | Method | FLOPs (P) ↓ | Speedup ↑ | Latency (s) ↓ | VBench ↑ | LPIPS ↓ | SSIM ↑ | PSNR ↑ |
> | :--- | :---: | :---: | :---: | :---: | :---: | :---: | :---: |
> | Wan 2.1 (T=50) | 8.32 | 1× | 364.13 | 0.7275 | - | - | - |
> | SemCache ($l$=3) | 2.79 | 2.81× | 129.58 | 0.6807 | 0.2167 | 0.7086 | 19.37 |
> | SemCache ($l$=6) | 2.84 | 2.76× | 131.93 | 0.6895 | 0.2074 | 0.7159 | 20.46 |
> | SemCache ($l$=9) | 2.91 | 2.72× | 133.87 | 0.7057 | 0.2013 | 0.7281 | 21.03 |
> | SemCache ($l$=12) | 3.07 | 2.69× | 135.36 | 0.7094 | 0.1939 | 0.7353 | 21.88 |
> | **SemCache ($l$=15)** | **3.12** | **2.66×** | **136.90** | **0.7123** | **0.1851** | **0.7513** | **22.97** |
> | SemCache ($l$=18) | 3.18 | 2.58× | 141.14 | 0.7116 | 0.1834 | 0.7484 | 22.42 |
>
>
> ## W4: The correlation between proxy-generated scores and LLM-generated scores
> In Appendix A.1, we visualize the comparative results between LLM and PSA. This includes a statistical analysis of semantic scoring using ChatGPT-4o to evaluate prompts from VBench across four dimensions: motion intensity, scene complexity, temporal dynamics, and visual dynamics. Additionally, (b) and (c) illustrate the semantic scoring variation trends during generation for 30 prompts containing dynamic, neutral, and static semantics on HunyuanVideo and Wan2.1, respectively.

---

### Official Review · Reviewer_pNyX · 2025-11-01

**Soundness:** 3
**Presentation:** 3
**Contribution:** 3
**Rating:** 6
**Confidence:** 3

**Summary:**

This paper proposes SemCache, a training-free and semantics-aware caching framework for accelerating video diffusion inference. The key idea is to perceive the semantic and motion dynamics of the input prompt to guide when and what intermediate features to cache. The method introduces two complementary modules: a Prompt Semantic-Aware (PSA) strategy that measures cross-attention input-output variations as a proxy for prompt-driven semantic complexity, and a Temporal Motion Metric (TMM) that quantifies inter-frame latent changes to identify motion-rich segments. Without retraining or calibration, SemCache achieves up to 2.66× inference acceleration on large diffusion transformers (HunyuanVideo, Wan2.1) while maintaining comparable video quality. The authors also validate the PSA and TMM components through ablations and compare against TeaCache and MagCache, showing superior trade-offs between speed and fidelity.

**Strengths:**

- The paper addresses a timely and important bottleneck: inference latency in video diffusion models. The observation that prompts with different semantic dynamics lead to varying redundancy in computations is intuitive and practically meaningful.

 - Compared to previous caching methods (which often rely purely on frame/step residual magnitudes or fixed schedules), the integration of a prompt semantic score to guide caching decisions is a noteworthy contribution.

 - The reported speedups (~2.45–2.66×) are non-trivial and the authors show evidence of maintaining output quality (VBench, PSNR/SSIM) across different models and scenarios.

 - The paper provides formulae for the PSA score and for the cumulative error threshold, which helps reproducibility and clarity.

**Weaknesses:**

1. While the paper describes when to reuse or recompute (via PSA + TMM) it does not clearly articulate what specific computations or layers are being cached/reused. For example: is caching happening at the level of transformer blocks, cross-attention layers, or entire time-step outputs? This ambiguity reduces reproducibility and clarity of the mechanism.

2. **Limited theory or validated correlation**: The PSA module uses cross-attention input/output magnitude and direction differences as a proxy for prompt semantic complexity. However the paper lacks deeper empirical or theoretical analysis showing that this proxy correlates strongly with “computational redundancy” across many prompts and models.

3. **Thresholding/hyper-parameters sensitivity**: The decision threshold based on PSA (and the choice of weights $\alpha, \beta$) appear somewhat heuristic. The paper has limited analysis of how sensitive the method is to these hyperparameters or how robust it is across different settings.

**Questions:**

1. How sensitive is the performance (both speedup and output quality) to the hyperparameters $\alpha, \beta$ and the threshold logic in Eq 8? Did you observe degradation when these change?

2. In very high-motion or highly dynamic scenes (or very long videos), does the caching mechanism still yield significant speedups? Are there failure cases where the overhead of evaluating PSA/TMM outweighs the caching benefit?

---

> ### Author Response · Authors · 2025-12-04
> **Response to Reviewer pNyX**
>
> ## W1: The core caching pipeline of SemCache.
> We sincerely thank the reviewers for their time and insightful feedback. We apologize for any confusion regarding SemCache.
> We clarify the complete pipeline of SemCache as follows:
> * **What is cached**: SemCache caches the model block outputs (specifically, the output-input difference $\Delta$) from fully computed timesteps, rather than caching raw intermediate features.
>
> * **When to cache**: The caching decision is dynamically determined by our PSA metric—when cumulative error $E_t < P_{st}$ (semantic threshold), we reuse cached outputs; otherwise, we perform full computation and update the cache.
>
> * **Where it operates**: Caching occurs at the Transformer block level, meaning the decision applies to complete block forward passes, not individual layers or attention heads.
> We have formalized this in Algorithm 1 (Appendix B) with explicit pseudocode showing the cache/compute decision logic.
>
> ## W2: Limited theory or validated correlation
> To better demonstrate that this proxy correlates strongly with computational redundancy, we conduct comparative experiments using 100 prompts for dynamic scenes,  and 100 prompts for static scenes, comparing a uniform acceleration strategy against SemCache. As shown in the table below, experimental results reveal that applying the same acceleration ratio to dynamic videos as to static videos leads to significant degradation in video generation quality.
>
> For static video generation, the model exhibits low sensitivity to caching errors, allowing more aggressive caching to reduce redundant computation. In contrast, for dynamic video generation, the model is highly sensitive to caching errors; excessive caching causes error accumulation, increasing the probability of artifacts in generated dynamic videos. SemCache dynamically perceives prompt semantics to achieve optimal trade-offs between acceleration and generation quality. These findings validate that this proxy correlates strongly with computational redundancy.
> | Method | Speedup $\uparrow$ | LPIPS $\downarrow$ | SSIM $\uparrow$ | PSNR $\uparrow$ |
> | :--- | :---: | :---: | :---: | :---: |
> | &nbsp;&nbsp;&nbsp;&nbsp;&nbsp;&nbsp;&nbsp;&nbsp;&nbsp;&nbsp;&nbsp;&nbsp;&nbsp;&nbsp;&nbsp;&nbsp;&nbsp;&nbsp;&nbsp;&nbsp;&nbsp;&nbsp;&nbsp;&nbsp;&nbsp;&nbsp;&nbsp;&nbsp;&nbsp;&nbsp;&nbsp;&nbsp;&nbsp;&nbsp;&nbsp;&nbsp; **Open-Sora 1.2 (51 frames, 480P)** | | | | |
> | SemCache (Static scenes w/o PSA) | 2.1$\times$ | 0.1715 | 0.7809 | 21.58 |
> | SemCache (Static scenes) | **2.1$\times$** | **0.1236** | **0.8206** | **25.84** |
> | SemCache (Dynamic scenes w/o PSA) | 2.1$\times$ | 0.2910 | 0.7103 | 17.16 |
> | SemCache (Dynamic scenes) | **2.03$\times$** | **0.1397** | **0.8128** | **23.39** |
> | &nbsp;&nbsp;&nbsp;&nbsp;&nbsp;&nbsp;&nbsp;&nbsp;&nbsp;&nbsp;&nbsp;&nbsp;&nbsp;&nbsp;&nbsp;&nbsp;&nbsp;&nbsp;&nbsp;&nbsp;&nbsp;&nbsp;&nbsp;&nbsp;&nbsp;&nbsp;&nbsp;&nbsp;&nbsp;&nbsp;&nbsp;&nbsp;&nbsp;&nbsp;&nbsp;&nbsp; **HunyuanVideo (129 frames, 1280$\times$720)** | | | | |
> | SemCache (Static scenes w/o PSA) | 2.45$\times$ | 0.195 | 0.7439 | 21.76 |
> | SemCache (Static scenes) | **2.45$\times$** | **0.1502** | **0.7819** | **24.98** |
> | SemCache (Dynamic scenes w/o PSA) | 2.45$\times$ | 0.2110 | 0.7118 | 18.22 |
> | SemCache (Dynamic scenes) | **2.38$\times$** | **0.1783** | **0.7405** | **21.06** |
> | &nbsp;&nbsp;&nbsp;&nbsp;&nbsp;&nbsp;&nbsp;&nbsp;&nbsp;&nbsp;&nbsp;&nbsp;&nbsp;&nbsp;&nbsp;&nbsp;&nbsp;&nbsp;&nbsp;&nbsp;&nbsp;&nbsp;&nbsp;&nbsp;&nbsp;&nbsp;&nbsp;&nbsp;&nbsp;&nbsp;&nbsp;&nbsp;&nbsp;&nbsp;&nbsp;&nbsp; **Wan 2.1 (81 frames, 832$\times$480)** | | | | |
> | SemCache (Static scenes w/o PSA) | 2.66$\times$ | 0.1763 | 0.7351 | 21.45 |
> | SemCache (Static scenes) | **2.66$\times$** | **0.1694** | **0.7604** | **23.78** |
> | SemCache (Dynamic scenes w/o PSA) | 2.66$\times$ | 0.2064 | 0.7252 | 17.06 |
> | SemCache (Dynamic scenes) | **2.49$\times$** | **0.1864** | **0.7481** | **21.76** |

---

> ### Author Response · Authors · 2025-12-04
> **Response to Reviewer pNyX**
>
> ## W3 and Q1: Thresholding/hyper-parameters sensitivity
> We thank the reviewer for this question.
>
> To analyze sensitivity, we conducted a fine-grained grid search for $\alpha \in [0, 1]$ (with $\beta=1-\alpha$) on both Wan 2.1 and HunyuanVideo. As illustrated in Fig. 5 in the revised manuscript and the table below, the default parameters were chosen based on objective optima to ensure the best efficiency-quality trade-off. VBench scores degrade gracefully around the optimal point, demonstrating robustness to parameter variations. Regarding the decision threshold ($p_{s_t}$), we emphasize that our threshold is dynamic (Equation 8) rather than static, adapting to the accumulated error at the current inference step.
>
>
> **Table: Ablation study of hyperparameter α on HunyuanVideo and Wan 2.1.**
>
> **(a) HunyuanVideo (129 frames, 1280×720)**
>
> | Method | α | Speedup | VBench Score |
> | :--- | :---: | :---: | :---: |
> | SemCache | 0.0 | 2.28× | 0.7543 |
> | SemCache | 0.2 | 2.21× | 0.7563 |
> | SemCache | 0.3 | 2.31× | 0.7554 |
> | **SemCache (Optimal)** | **0.4** | **2.66×** | **0.7598** |
> | SemCache | 0.5 | 2.39× | 0.7547 |
> | SemCache | 0.6 | 2.42× | 0.7541 |
> | SemCache | 0.7 | 2.36× | 0.7588 |
> | SemCache | 0.8 | 2.44× | 0.7539 |
> | SemCache | 0.9 | 2.36× | 0.7581 |
> | SemCache | 1.0 | 1.93× | 0.7556 |
>
> **(b) Wan 2.1 (81 frames, 832×480)**
>
> | Method | α | Speedup | VBench Score |
> | :--- | :---: | :---: | :---: |
> | SemCache | 0.0 | 2.41× | 0.7119 |
> | **SemCache (Optimal)** | **0.2** | **2.66×** | **0.7123** |
> | SemCache | 0.3 | 2.51× | 0.7102 |
> | SemCache | 0.4 | 2.49× | 0.7115 |
> | SemCache | 0.5 | 2.59× | 0.7113 |
> | SemCache | 0.6 | 2.62× | 0.7120 |
> | SemCache | 0.7 | 2.60× | 0.7119 |
> | SemCache | 0.8 | 2.64× | 0.7120 |
> | SemCache | 0.9 | 2.63× | 0.7117 |
> | SemCache | 1.0 | 2.40× | 0.7106 |
>
> ## Q2: Inference efficiency of SemCache at different video lengths and resolutions
> We appreciate the reviewer's attention to the performance of our method in long video generation and the practical overhead of our method.
>
> To assess the scalability of our method across different video configurations, we evaluate performance on varying resolutions and frame counts. As shown in Fig. 6 in the revised manuscript, our method achieves consistent acceleration across all tested settings, delivering approximately 2.2-2.5× speedup on a single GPU and 3.0-3.3× speedup on dual GPUs. The stable performance across different resolutions (544 × 960 to 720 × 1280) and frame counts (129 to 192 frames) demonstrates the method's robustness. We provide comprehensive profiling results below to demonstrate that PSA and TMM introduce negligible computational costs relative to the substantial speedup achieved.
>
> **PSA+TMM Computational Overhead (50 timesteps)**
>
> | Model | Resolution | Frames | Total Overhead | Baseline Generation Time | Relative Overhead |
> |:------|:-----------|:------:|:--------------|:------------------------|:-----------------|
> | Open-Sora 1.2 | 848x480 | 51 | 125 ms | 1,490 ms | 8.4% |
> | Wan 2.1 | 832×480 | 81 | 170 ms | 7,280 ms | 2.3% |
> | HunyuanVideo | 1280×720 | 129 | 240 ms | 52,000 ms | 0.46% |

---

### Meta-Review · Area_Chair_WhXt · 2026-01-04

**Summary:**

The submission initially received relatively negative comments. The main concerns can be summarized into the following points:
1. The presentation needs to be further improved, many important details are missing. For example, the details for integrating TMM and PSA, implementation details in the experimental section, threshold decision, and the selection of specific cached layers.
2.  More comprehensive experimental results over different prompts to validate the claims in the introduction section, such as results on different prompts.
3. The connection between cross-attention variation and semantic complexity lacks a rigorous analytical foundation or sensitivity analysis.
4. Some figures and equations are confusing.

In the rebuttal, the authors have addressed some of the concerns by providing extra results on more comprehensive ablations. However, the main weakness of the submission is the presentation, as many important logics are not well-explained and details are missing. Thus, I recommended Reject.

**Reviewer Concerns:**

Some concerns about more comprehensive ablations have been addressed, the bad presentation and missing rigorous analytical analysis are not well-addressed.

**Reviewer Scores:**

I think all reviewers will maintain their original ratings.

---

### Decision · Program_Chairs · 2026-01-26

Reject